# Sirtuins Contribute to the Migraine–Stroke Connection

**DOI:** 10.3390/ijms26146634

**Published:** 2025-07-10

**Authors:** Jan Krekora, Michal Fila, Maria Mitus-Kenig, Elzbieta Pawlowska, Justyna Ciupinska, Janusz Blasiak

**Affiliations:** 12nd Department of Cardiology, Medical University of Lodz, 92-213 Lodz, Poland; ejkrekora@op.pl; 2Department of Developmental Neurology and Epileptology, Polish Mother’s Memorial Hospital Research Institute, 93-338 Lodz, Poland; michal.fila@iczmp.edu.pl; 3Department of Periodontology, Preventive Dentistry and Oral Pathology, Jagiellonian University, 31-532 Krakow, Poland; maria.mitus-kenig@uj.edu.pl; 4Department of Pediatric Dentistry, Medical University of Lodz, 92-217 Lodz, Poland; elzbieta.pawlowska@umed.lodz.pl; 5Clinical Department of Infectious Diseases and Hepatology, H. Bieganski Hospital, 91-347 Lodz, Poland; jciupinska@bieganski.com.pl; 6Faculty of Medicine, Mazovian University in Plock, 09-240 Plock, Poland

**Keywords:** migraine, stroke, sirtuins, migraine–stroke connection, oxidative stress

## Abstract

The prevalence of stroke in patients with migraine is higher than in the general population, suggesting certain shared mechanisms of pathogenesis. Migrainous infarction is a pronounced example of the migraine–stroke connection. Some cases of migraine with aura may be misdiagnosed as stroke, with subsequent mistreatment. Therefore, it is important to identify these shared mechanisms of pathogenesis contributing to the migraine–stroke connection to improve diagnosis and treatment. Sirtuins (SIRTs) are a seven-member family of NAD^+^-dependent histone deacetylases that can epigenetically regulate gene expression. Sirtuins possess antioxidant properties, making them a first-line defense against oxidative stress, which is important in the pathogenesis of migraine and stroke. Mitochondrial localization of SIRT2, SIRT3, and SIRT4 supports this function, as most reactive oxygen and nitrogen species are produced in mitochondria. In this narrative review, we present arguments that sirtuins may link migraine with stroke through their involvement in antioxidant defense, mitochondrial quality control, neuroinflammation, and autophagy. We also indicate mediators of this involvement that can be, along with sirtuins, therapeutic targets to ameliorate migraine and prevent stroke.

## 1. Introduction

Migraine and ischemic stroke, hereafter referred to as stroke unless stated otherwise, are serious neurological diseases that significantly contribute to disability, yet their long-term outcomes can be markedly different. While migraine may lead to considerable changes in personal and professional life, it is generally regarded as a benign condition. In contrast, stroke is a life-threatening disease and one of the leading causes of death and long-term disability worldwide [1].

Migraine is a highly heterogeneous group of syndromes, some of which are characterized by the absence of headaches (“silent migraine”). In about one-third of cases, migraine is marked by the occurrence of visual, sensory, or speech symptoms called migraine aura (MA), which can be mistaken for a stroke and lead to unnecessary acute or secondary prevention treatments [2]. On the other hand, such misdiagnosis may lead to the mistreatment of stroke patients as if they were migraine cases, particularly in instances of chronic migraine. Migraine without aura (MO) is more common among migraine cases, but its connection to stroke is not as evident as that in MA [3]. However, a genome-wide association study (GWAS) demonstrated that MO was associated with stroke even more strongly than MA [4].

It is difficult to estimate the prevalence of stroke among MA and MO patients, as most studies include only MA patients, and some do not distinguish between MA and MO. A 2005 meta-analysis found that the relative risk of stroke in MA or MO patients was 2.16 or 1.83, respectively [5]. A large US-based study showed that women with probable migraine with visual aura had 1.5 times greater odds of ischemic stroke compared to women without migraine [6]. That study did not find any difference in stroke risk for MO women and their counterparts without migraine. These and other studies suggest that the prevalence of stroke in MO patients may be similar to that in the general population. Despite some studies reporting statistical significance, their biological and medical relevance may be negligible.

Migraine-related cerebral infarction, a syndrome exemplifying the migraine–stroke connection, is characterized by the occurrence of stroke during migraine, in both its ictal and interictal phases [7]. Migrainous infarction is associated with structural changes in the brain affected by migraine [8].

Although some authors highlight common aspects of pathophysiology, risk factors, and clinical outcomes, the mechanistic relationship between migraine and stroke remains not entirely understood (reviewed in [9]). In addition to smoking, obesity, and hypertension—which are linked to the development of many diseases—migraine and stroke also share several other, more specific risk factors, such as cortical spreading depolarization or depression (CSD), atrial fibrillation, patent foramen ovale, and some mitochondrial diseases. Some of these factors are more strongly associated with stroke in MA patients than in their MO counterparts or are significant only for MA patients. Surely, the migraine–stroke connection is not limited to shared risk factors, but also includes mechanisms of pathogenesis, clinical presentation, and treatment (reviewed in [10]).

Sirtuins are nicotinamide adenine dinucleotide (NAD^+^)-dependent histone deacetylases that may epigenetically regulate gene expression and contribute to the pathogenesis of neurological diseases (reviewed in [11]). They exhibit antioxidant properties, which are crucial for understanding the migraine–stroke connection, as oxidative stress plays a significant role in the pathogenesis of both diseases [12,13]. Sirtuins also regulate additional effects related to oxidative stress and may influence migraine and stroke, including apoptosis, autophagy, and neuroinflammation [14]. Therefore, sirtuins are potential links between the pathogenesis of migraine and stroke.

In this narrative review, we discuss the migraine–stroke connection, provide a basic characterization of sirtuins, and explore their role in the pathogenesis of both migraine and stroke. We highlight the effects relevant to the shared pathogenesis of migraine and stroke, including their mediators, along with a therapeutic perspective aimed at targeting sirtuins and other proteins involved in sirtuin signaling to alleviate migraine and prevent stroke.

## 2. The Migraine–Stroke Connection

The relationship between migraine and stroke includes many facets, yet few can be mechanistically explained.

### 2.1. Epidemiology

The prevalence of stroke among MA patients is roughly twice as high as that in MO patients or individuals without migraine [15,16]. The risk of stroke in MO patients is not well established. Despite the high risk in MA patients, the prevalence of stroke among migraine patients remains low. A 2020 study analyzing data from 834,875 young adults in the US (ages 18–44) with migraines indicated that the prevalence of stroke was 1.3%, significantly higher in MA patients at 3.7% compared to 1.2% in MO patients [15]. However, the population-based cohort study of middle-aged and elderly individuals, known as the Rotterdam Study, showed that having migraines was linked to a higher risk of incident stroke; however, this association was not statistically significant [17].

The Northern Manhattan Study (NOMAS) is a population-based, multi-ethnic cohort study of stroke incidence and risk factors (https://northernmanhattanstudy.org/, accessed on 15 June 2025). Participants in NOMAS were assessed for migraine symptoms according to ICH-2 [18]. No association was found between MA or MO and the risk of either stroke or combined cardiovascular events. However, migraine was linked to a heightened risk of stroke among active smokers, but not among nonsmokers.

The prevalence of migraine is higher in women than in men, with rates reaching up to threefold [19]. This sexual dimorphism in migraine prevalence alters the understanding of its association with stroke when accounted for by sex. In the Women’s Health Study, the risk of stroke among MA women was higher, even after adjusting for additional stroke risk factors [20,21]. The risk of stroke in women with migraine may depend on sex-specific factors, such as oral contraceptive use, and has also been reported to increase in women with MO [21]. Additionally, pregnant women exhibited an increased risk of stroke with both MA and MO [22]. Stroke is also characterized by sexual dimorphism. While males have a higher incidence of stroke throughout most of their lifespan, older females carry a greater burden of the condition [23]. Differences in stroke between sexes arise from various factors, including those inherent to the sex chromosomes and the influence of lifelong sex hormone exposure. However, further studies are needed to determine whether these differences are due to the high baseline prevalence of migraine rather than a direct causal association [24].

### 2.2. Shared Risk Factors

Smoking and obesity are regarded as some of the most serious modifiable risk factors for migraine and stroke [7]. However, they play a role in the development of numerous disorders, making it challenging to specifically and mechanistically link them to shared risk factors for migraine and stroke. Another risk factor that can be at least partially modified is high blood pressure [25]. Other common risk factors for migraine and stroke include excessive vasodilation, CSD, cerebral hypoperfusion, oligemia, endothelial dysfunction or structural vasculopathy, atrial fibrillation, patent foramen ovale, microembolism, hypercoagulability, mitochondrial encephalomyopathy, lactic acidosis, and stroke-like episodes (MELAS), and inflammation (reviewed in [8]). Some relate only to MA, while others will be discussed in greater detail in later sections.

### 2.3. Cortical Spreading Depolarization/Depression

Cortical spreading depolarization refers to a near-complete depolarization of neurons and glial cells that may propagate slowly at a speed of 3 to 5 mm/min across the brain cortex [26]. An expansion of the depolarization state leads to the depression of electrocortical signals, CSD. This phenomenon is most accurately described in the occipital lobe, though it can also occur in other areas of the cortex, leading to visual, sensory, motor, and language auras in migraine [27]. However, CSD and CGRP function at the intersection of the vasculature and cortical neurons, thereby contributing to the overall pathogenesis of migraine, including not just aura [28].

The fact that CSD is linked to migraine aura and that MA patients are more likely to experience a stroke than MO patients and the general population suggests that CSD may play a common role in the pathogenesis of both MA and stroke.

CSD may lead to temporary vasoconstriction, disrupt the integrity of the blood–brain barrier, reduce cerebral blood flow, and induce endothelial dysfunction, which could potentially heighten ischemic vulnerability and contribute to stroke [8]. CSD is also thought to contribute to migrainous infarction [5].

### 2.4. Migrainous Infarction and Brain Structural Alterations

In MA patients, the International Classification of Headache Disorders (ICHD-3) specifies migrainous infarction, which occurs during migraine attacks, and non-migrainous infarction, which occurs in the interictal phase [29]. Migrainous infarction is classified as a complication of migraine and is described as “One or more migraine aura symptoms associated with an ischemic brain lesion in the appropriate territory demonstrated by neuroimaging”. Typical imaging features in chronic migraine (CM) on magnetic resonance imaging (MRI) include deep white matter T2 hyperintensities, infratentorial hyperintensities, and infarct-like lesions, typically in the posterior circulation, in the context of migraine with aura [30].

Migrainous infarction exemplifies the consequences of structural changes in migraine. However, not all brain structural alterations associated with migraine should be attributed to migrainous infarction. Some of these changes may predispose individuals to stroke or reflect ongoing stroke pathogenesis. The structural changes associated with migraine can be categorized into several groups: white matter hyperintensities/white matter abnormalities, silent infarct-like lesions, ischemic lesions, and volumetric changes in both gray and white matter [31]. Silent infarct-like lesions in migraine patients are associated with a higher risk of stroke [16,18,32].

### 2.5. Genetics/Epigenetics

A significant number of monogenic vascular diseases present migraine or migraine-like symptoms within their phenotypic range. The most notable example is likely CADASIL, which is caused by a missense mutation in the notch 3 receptor (*NOTCH3*) gene, encoding a transmembrane receptor protein expressed on vascular smooth muscle cells [33]. CADASIL is the most common form of monogenic adult-onset stroke [4]. Mitochondrial myopathy with encephalopathy, lactic acidosis, and stroke-like episodes (MELAS) is often viewed as a combination of migraine and stroke [34]. Headaches associated with nausea and vomiting may occur in MELAS [35]. Genetically, MELAS is characterized by a pathogenic nucleotide variant in mitochondrial DNA, with the most common being m.3243A>G, which occurs in the mitochondrially encoded TRNA-Leu (UUA/G) 1 (*MTL1*) gene [36]. However, migraine cannot be regarded as a monosymptomatic form of MELAS because no association has been found between mutations typical of MELAS and migraine (reviewed in [37]). Familial hemiplegic migraine (FHM) is a rare monogenic disease characterized by migraine as a predominant phenotypic feature. It is considered the monogenic form of migraine, although the disease is genetically heterogeneous. FHM can occur in one of three subtypes: FHM1, FHM2, and FHM3, with underlying pathogenic nucleotide variants in the calcium voltage-gated channel subunit alpha 1 A (*CACNA1A*), ATPase Na+/K+ transporting subunit alpha 2 (*ATP1A2*), and sodium voltage-gated channel alpha subunit 1 (*SCN1A*) genes, respectively [38,39,40]. All these genes are involved in ion transport within the CNS. A transgenic mouse model of FHM1 demonstrated increased susceptibility to stroke, leading to the conclusion that enhanced vulnerability to CSD may predispose migraine patients to stroke.

Studies on candidate genes and genome-wide association studies have led to the identification of several genes relevant to the link between migraine and stroke. These include the methylenetetrahydrofolate reductase (*MTHFR*) and angiotensin I converting enzyme (*ACE*) genes, which have been reported to generally associate migraine with stroke [41,42,43,44]. On the other hand, the 9p21 region, leiomodin 2 (*LMOD2*), and WASP-like actin nucleation-promoting factor (*WASL2*) genes have been reported to associate MO with large-artery stroke [45]. Therefore, genetic studies suggest that both MA and MO may be linked to stroke, as clearly demonstrated in the cases of cervical artery dissection, which is a comorbidity of migraine. An analysis using GWAS data sets for stroke (METASTROKE) and migraine (IHGC) indicated a substantial genetic overlap between stroke and migraine, with stronger effects noted between MO and stroke compared to MA and stroke [45].

Several other vascular diseases related to stroke, along with symptomatic migraine and genetic factors, have been described. Consequently, many genes may be relevant to the migraine–stroke connection (reviewed in [4]).

Epigenetic studies suggest a crucial role for all three primary epigenetic mechanisms: DNA methylation, post-translational histone modifications, and the action of non-coding RNAs, particularly microRNAs, in both migraine and stroke [46,47]. Still, there are no studies aimed at determining the role of epigenetic mechanisms as a shared element in migraine and stroke pathogenesis.

In summary, there are several common aspects of migraine and stroke that may reflect shared mechanisms of pathogenesis for both diseases. Many risk factors are common to migraine and stroke; some are illustrated in Figure 1. However, they exhibit different specificities for each disease and generally act as risk factors for other syndromes. The common denominator for most, if not all, of these risk factors is oxidative stress, and elucidating the mechanistic connection between a risk factor and the migraine/stroke phenotype may contribute to better prevention and therapy for both diseases.

## 3. Sirtuins

In mammals, sirtuins (silent information regulators, SIRTs) are a family of class III histone deacetylases consisting of seven members: SIRT1 to SIRT7. All SIRTs contain a nicotinamide adenine dinucleotide (NAD^+^)-binding catalytic domain [48]. They are characterized by various localizations, including the nucleus, cytoplasm, and mitochondria (Table 1). In most cell types, SIRT7 is primarily localized in the nucleolus, making it a unique member of the SIRT family [49].

Sirtuins perform numerous functions that are significant in both physiology and pathology. While some of these functions are listed in Table 1, most, if not all, may be directly or indirectly linked to the involvement of SIRTs in gene expression regulation, primarily through modifications of chromatin proteins and transcription factors [61]. Table 1 shows only the main localizations, activities, and targets; more details can be found in some excellent reviews, e.g., [52,62]. Emerging evidence demonstrates an important role of sirtuins in the pathogenesis of various diseases, including cardiovascular disease, cancer, neurodegenerative diseases, epilepsy, liver diseases, premature aging, and other serious syndromes [54]. Sirtuins are also implicated in pain signaling and may play a significant role in pain chronification [63].

The role of sirtuins in physiological functions and pathological effects is underscored by several molecular mechanisms that do not directly relate to SIRTs’ regulation of gene expression. These mechanisms encompass DNA repair, autophagy, cell proliferation and death, mitochondrial quality control, energy metabolism, and additional effects that are still under investigation [64,65,66].

The involvement of sirtuins in antioxidant and redox signaling is crucial since oxidative stress is linked to the pathogenesis of numerous syndromes. SIRT2 has been reported to deacetylate nuclear factor erythroid 2-related factor 2 (NRF2), which regulates the expression of various genes involved in antioxidant defense [67]. However, the antioxidant role of sirtuins should not be overgeneralized, as increased levels of SIRT4 have been shown to be associated with elevated levels of RONS. A study on mouse mitochondria demonstrated that the upregulation and downregulation of SIRT4 increased and decreased RONS production, respectively, and affected the mitochondria [68]. This study highlights the complex role of sirtuins in redox homeostasis; however, many studies consistently demonstrate an antioxidant potential for this protein family [69]. Bioinformatic analyses reveal a complex network of interactions among sirtuins and various proteins, as well as between members of the sirtuin family, which may explain sirtuin involvement in antioxidant and redox signaling [70].

Sirtuins exert their effects throughout the entire organism, including the central and peripheral nervous systems. Furthermore, their involvement in neuroinflammation, regulation of brain function, oxidative stress, and mitochondrial quality control highlights their significant potential in maintaining nervous system homeostasis. Thus, sirtuins may play a critical role in the pathogenesis of and therapeutic strategies for neurological and psychiatric diseases. Additionally, the influence of sirtuins on learning and memory is consistently reported [71]. Sirtuins enhance cognition by improving synaptic plasticity, regulating epigenetic processes, and engaging in molecular pathways linked to oxidative stress that affect mitochondrial function [72].

The involvement of sirtuins in the pathogenesis of neurodegenerative diseases, including Alzheimer’s disease (AD), Parkinson’s disease (PD), and Huntington’s disease (HD), is well documented [73]. That involvement is based on the specific localization of sirtuins and their ability to cross the blood–brain barrier. For example, SIRT1 can be found in the prefrontal complex and the hippocampus, which are important in AD pathogenesis [74]. Further information about the role of sirtuins in neurodegenerative diseases can be found elsewhere [46].

Sirtuins may play a crucial role in mechanical allodynia, hyperalgesia, and recovery following nerve injury by improving mitochondrial function, decreasing oxidative stress, and reducing neuroinflammation [75,76]. These effects may play a crucial role in the pathogenesis of migraine and ischemic stroke [77,78].

## 4. Sirtuins in Migraine

Although oxidative stress is potentially implicated in the pathogenesis of many syndromes, its role in migraine is logical and can be justified by at least three aspects. First, the migraine-affected brain requires and produces more energy than normal, and this overproduction is connected to the excess generation of reactive oxygen and nitrogen species (RONS) by brain mitochondria, resulting in oxidative stress [79]. Second, numerous migraine triggers are directly or indirectly associated with oxidative stress [80]. There are some examples of dietary triggers: (1) Alcohol is partially metabolized by cytochrome P450-2E1, producing byproducts such as superoxide and hydrogen peroxide [81]. (2) Water deprivation raises the levels of arginine vasopressin, which stimulates the release of endothelin; this compound, acting on its receptor, releases superoxide anion [82]. (3) Monosodium glutamate may increase brain glutamate levels, thereby activating ionotropic and metabotropic glutamate receptors on neurons, which leads to calcium influx and higher rates of oxidative phosphorylation and oxidant production [83]. (4) Aspartame, which contains a methyl ester linkage, is metabolized into methanol, then converted into formaldehyde and subsequently to formate molecules that inhibit mitochondrial complex III, resulting in the release of superoxide, peroxyl, and hydroxyl radicals [84]. (5) Tyramine, present in red wine and aged cheese, is broken down by monoamine oxidase types A and B, producing hydrogen peroxide as a byproduct [85]. The significance of oxidative stress in other migraine triggers—including environmental, physiological, behavioral, and pharmacological factors—can be found in Borkum’s excellent review [86]. Next, RONS might show a pro-nociceptive action [87]. Therefore, the overproduction of RONS, associated with brain mitochondria being overloaded due to excessive energy production, may be critical for migraine induction in a sensitive brain and contribute to the disease’s development, including its chronification. Consequently, effective mitochondrial quality control (mtQC) and a functional antioxidant system may be essential for preventing migraine attacks and headache chronification.

The primary components of the cellular antioxidant system include antioxidant enzymes, DNA repair proteins, and low molecular-weight antioxidants [88]. Most RONS related to migraine are produced in the brain’s mitochondria; therefore, the mitochondrial localization of the antioxidant system’s components may be essential for its efficacy, as most RONS are characterized by a short lifetime [89]. As presented in Table 1, SIRT3, SIRT4, and SIRT5 are localized to mitochondria. Most, if not all, sirtuins may mitigate oxidative stress and play a role in the DNA damage response (DDR), whose main element is DNA repair. Therefore, the basic properties of sirtuins predispose these proteins to mitigate mitochondria-borne oxidative stress. Consequently, they may play a crucial role in protecting against the induction and development of migraine. As stated by Borkum, migraine may be the reaction of a sensitive brain against oxidative stress associated with increasing levels of RONS [90]. Therefore, migraines may be triggered if the brain’s antioxidant system is compromised. Decreased activity of catalase, an enzyme that deactivates hydrogen peroxide, has been demonstrated in the serum of patients with a family history of migraine, suggesting that migraine may be causally linked to disturbances in the antioxidant system [91]. Therefore, decreased expression and reduced functionality of sirtuins may be linked to migraine occurrence and progression.

In this context, it is interesting to ask whether the prevalence of migraine is higher in patients with other syndromes characterized by catalase dysfunction. Acatalasemia is a syndrome marked by a genetic deficiency of catalase, and there are no studies on migraine prevalence in acatalasemia [92]. Other research on syndromes with dysfunctional catalase generally considers it within the broader framework of antioxidant defense. As a result, it is difficult to isolate this condition from other aspects of antioxidant defense, and that is why, at present, this question cannot be answered reliably.

As noted in the previous section, SIRT2 may influence antioxidant capacity by deacetylating NRF2 [63]. It was observed that the expression of both NRF2 and SIRT2 decreased in the rat model of neuropathic pain [80]. Although migraine is not typically classified as a neuropathic pain, emerging evidence suggests that it may be linked to damage in various brain structures [93,94]. Therefore, neuropathic pain and migraine may share some common pathways in their pathogenesis, potentially involving the interaction between SIRT2 and NRF2. As suggested, the deacetylation of NRF2 by SIRT2 might reduce the expression of the NAD(P)H quinone dehydrogenase 1 (*NQO1*) gene, which is a target gene for NRF2. NQO1 plays a vital role in various aspects of cellular functioning, including antioxidant defense, and may thus link migraine with oxidative stress through SIRT2. The upregulation of *SIRT2* restores the expression of the *NFE2L2* gene, which encodes NRF2 and NQO1, while also decreasing oxidative stress in the injured spinal cord of rats, which leads to improvements in thermal hyperalgesia and mechanical allodynia, highlighting the potential of SIRT2 in migraine. Additionally, SIRT1 has been shown to play a protective role against chronic pain through various mechanisms, including the regulation of neuroinflammation, reducing oxidative stress, and improving mitochondrial dysfunctions [59].

The protective role of SIRT1 in migraine was demonstrated in a study in which miRNA-155-5p was found to promote neuroinflammation and central sensitization in the trigeminal nucleus caudalis of a nitroglycerin-induced mouse model of CM [95]. The miR-155-5p agomir increased CGRP and c-Fos expression, as well as microglial activation, which SRT1720, a SIRT1 activator, subsequently alleviated. miR-155 has been reported to be involved in regulating pain thresholds and proinflammatory cytokines in several pain models [96]. We have no consistent information suggesting that a weak effect of SRT1720 on sirtuins other than SIRT1 would not be an off-target effect. Consequently, the activation of SIRT1 reduced the worsening of neuroinflammation and central sensitization caused by a miR-155-5p agomir. The expression of miR-155 in the monocytes of migraine patients was linked to phenotype, disease severity, and the inflammatory profile [97]. The highest expression of miR-155 was observed in patients with chronic migraine and medication overuse. These studies indicate that the pro-migraine effect of miRNA-155-5p is inhibited by SIRT1, suggesting that SIRT1 may play a protective role in migraine pathogenesis. Overall, the development of migraine may be associated with downregulated levels of SIRT1.

SIRT1 has also been shown to support migraine therapy with ligustrazine, the main component of the traditional edible-medicinal herb *Ligusticum chuanxiong* Hort., which is characterized by numerous clinical effects, including the expansion of small arteries [98]. Ligustrazine reduced migraine-like behaviors in NTG-induced mice and improved neuroinflammation and associated oxidative damage in brain tissue. However, co-treatment with the SIRT1 inhibitor EX527 negated the protective effects of LGZ. Mechanistically, LGZ reduced neuroinflammation by upregulating SIRT1 expression and subsequently inhibiting the activation of the NF-κB pathway in microglia. LGZ significantly preserved the stability of SIRT1 in microglia. The paracrine interaction between the NTG model and the LPS/IFNG-stimulated microglial cell model for neuroinflammation showed that culture medium from LPS/IFN-γ-treated microglia aggravated neuronal damage and oxidative stress but was suppressed by treating LPS/IFN-γ-induced microglia with ligustrazine. This effect was attributed to the activation of NRF2 signals in neurons. Therefore, SIRT1 may be a pharmacological target of ligustrazine, which reduces migraine-associated neuroinflammation and oxidative stress by interfering with the crosstalk between microglia and neurons.

SIRT7 regulates mitochondrial biogenesis and function by modulating the expression of nuclear-encoded mitochondrial genes, and SIRT7 deficiency leads to mitochondrial dysfunction, including altered morphology, reduced oxidative phosphorylation, and increased RONS production [99]. Moreover, SIRT7 plays a role in the mitochondrial unfolded protein response, a key mechanism to maintain mitochondrial proteostasis [100]. Since mitochondrial dysfunctions can be involved in the pathogenesis of cardiac myopathy and sirtuins play a role in mitochondrial quality control, it is reasonable to explore the role of sirtuins in cardiac myopathy. Repeated infusions of inflammatory soup in rats resulted in cutaneous hyperalgesia, and the SIRT1 activator SRT1720 alleviated the hyperalgesia [101]. Furthermore, activation of SIRT1 increased the expression of peroxisome proliferator-activated receptor gamma coactivator 1-alpha (PGC-1α), transcription factor A (TFAM), and nuclear respiratory factors 1 and 2 (NRF1 and NRF2), while enhancing the ATP content and mitochondrial membrane potential. Therefore, SIRT1 may exert a protective effect in migraine by improving mitochondrial functions.

To further explore the role of SIRT1 in chronic migraine (CM), the effect of reactive oxygen and nitrogen species (RONS)-mediated central sensitization in CM was investigated using an ischemia-stroke (IS)-based rat model of CM [102]. A reduced expression of SIRT1 and increased levels of RONS were found in the trigeminal nucleus caudalis of CM animals. Treatment with an RONS scavenger, a SIRT1 activator, and an inhibitor of mitochondrial fission relieved allodynia and decreased the increase in CGRP, mitogen-activated protein kinase 1 (ERK), and glutamate receptor ionotropic, NMDA 2B (NMDAR2B) phosphorylation, which are crucial in migraine pathogenesis [103,104,105]. The SIRT1 activator and inhibitor decreased and increased RONS levels, respectively, and the regulatory effect of SIRT1 on RONS might be mediated by the mitochondrial fission protein DRP1. Overall, these results highlight the importance of SIRT1 in chronic migraine (CM) through its role in regulating the production of RONS, which are involved in modulating central sensitization in CM.

A summary of the potential and reported involvement of SIRT1 and SIRT2 is presented in Figure 2. Oxidative stress and RONS may be produced by the migraine-affected brain and/or may result from various migraine triggers. RONS may contribute to neuroinflammation, central sensitization, and migraine chronification. SIRT2 may deacetylate NRF2, which regulates many antioxidant proteins. The deacetylation of NRF2 by SIRT2 might decrease the expression of the NQO1 gene, which is a target gene for NRF2. NQO1 plays an important role in various aspects of cellular function, including antioxidant defense. SIRT1 may inhibit BDNF, which otherwise promotes oxidative stress, neuroinflammation, and central sensitization. Additionally, SIRT1 may enhance mtQC by increasing the expression of PGC-1α, TFAM, and NRF1/2, thereby increasing ATP content and mitochondrial membrane potential. There is no direct evidence regarding the involvement of sirtuins other than SIRT1 or SIRT2 in migraine, but reports on the role of other sirtuins in neurological disorders justify further research on this subject [106].

The connection between migraine and stroke, along with the involvement of sirtuins in pain transmission, raises the question of the potential role these proteins play in stroke pathogenesis.

## 5. Sirtuins in Stroke

There are many reports on the involvement of sirtuins in stroke pathogenesis and rehabilitation, but this section will only present studies related to oxidative stress, autophagy, and neuroinflammation. This focus aims to facilitate the comparison of sirtuin-related effects in stroke and migraine. While human migraine studies can be conducted during both the ictal and interictal phases, many stroke-related studies, for obvious reasons, are primarily carried out in the post-stroke period.

### 5.1. Oxidative Stress

In a manner similar to migraines, oxidative stress is frequently reported to be involved in the pathogenesis of strokes. It is regarded as a factor associated with stroke as the prevalence of strokes increases and the effectiveness of the antioxidant system diminishes with age [10]. However, this argument can be applied to many other syndromes, but migraine does not fit that pattern.

In hypoxia, electrons generated in the ETC can interact with transition metals and produce RONS, as they cannot all combine with oxygen and be reduced to water [107]. Restoration of blood flow intensity leads to a sharp increase in oxygen concentration, which facilitates its interaction with electrons and the production of peroxide. After ischemia/reperfusion, reactive oxygen and nitrogen species (RONS) are generated, causing effects such as tissue acidification, changes in membrane potential, and calcium influx, among others [108]. Therefore, two elements related to oxidative stress may play an important role in stroke pathogenesis: mitochondria, the primary source of RONS, and the cellular antioxidant system. As specified in the previous section, sirtuins might improve mtQC and enhance the expression of antioxidant proteins (Figure 2). However, while oxidative stress and RONS may be common denominators in migraine and stroke, the underlying mechanisms differ in several respects. RONS in the brain may affect cerebral blood flow, inducing vasodilation, damaging endothelial cells, and increasing the permeability of the blood–brain barrier (BBB), all of which are crucial in stroke pathogenesis [109]. Oxidative stress was also monitored throughout the rehabilitation process in post-stroke patients [110]. The systemic oxidative status and progress in rehabilitation outcomes were evaluated before and after a six-week rehabilitation treatment. A decrease in serum levels of hydroperoxide and an improvement in relative antioxidant capacity were observed following rehabilitation. Moreover, a partial negative correlation was found between the walk test and hydroperoxide, as well as a positive correlation between the test and relative antioxidant capacity during the rehabilitation period. Therefore, parameters of oxidative stress may serve as supplementary indices in monitoring the progress of rehabilitation in post-stroke patients. Further studies on this subject could assess the suitability of these parameters in predicting the chances of recovery.

Several studies have demonstrated the importance of sirtuins in oxidative stress and RONS-related stroke pathogenesis. A decrease in SIRT3 levels was observed in mouse neurons that experienced oxygen-glucose deprivation, mimicking a stroke, followed by reperfusion [111]. Additionally, the translocase of the outer mitochondrial membrane 20 (TOMM20) decreased after reperfusion, indicating the activation of mitophagy, a specialized form of autophagy that targets damaged and dysfunctional mitochondria. At 4 h after reperfusion, the cleavage of caspase-3 increased, suggesting apoptosis. These results suggest that SIRT3 may play a crucial role in neurons during and after a stroke, likely through its involvement in mtQC.

The upregulation of SIRT3 and its downstream substrates, forkhead box O3a (FOXO3A) and superoxide dismutase 2 (SOD2), a key enzyme in the antioxidant defense, was observed in a mouse model of transient middle cerebral artery occlusion (MCAO) treated with the ketones beta-hydroxybutyrate and acetoacetate [112]. Ketone treatment led to improved neurological functions, assessed through the neurological score and performance in open-field tests. SIRT3 directly deacetylates SOD2 at specific lysines K53, K68, and K89, leading to its activation and improving its ability to neutralize mitochondrial superoxide radicals [113]. Another activator of SOD2 is FOXO3A, which binds to the promoter region of the SOD2 gene, thereby increasing its transcription [114]. Therefore, the SIRT3–FOXO3A–SOD2 pathway may be considered for stroke therapy and prevention with the involvement of ketones.

In a case-control study, patients with acute stroke showed a lower plasma level of SIRT1 than the controls [115]. Furthermore, there is a negative correlation between SIRT1 levels and stroke scores, suggesting that SIRT1 may be predictive of stroke severity.

### 5.2. Autophagy

Sirtuins are reported to modulate autophagy, a process through which cells self-digest and recycle damaged and/or no longer needed components—“cellular waste” [116]. However, the functions of autophagy extend beyond merely removing and recycling unnecessary materials, as it plays a role in various physiological and pathological processes, including those associated with the pathogenesis of numerous diseases [117]. Most reports regarding the role of sirtuins in autophagy regulation emphasize SIRT1 and SIRT6, while SIRT5 and SIRT7 are considered to have lesser significance (reviewed in [51]). Sirtuins may regulate the expression of autophagy-related genes and post-translationally modify autophagy proteins. Additionally, sirtuins can tag proteins for degradation. Autophagy can be activated in response to stroke in various types of cells present in the damaged brain area, including neurons, glial cells, and microvascular cells [118]. Autophagy, by definition, is a process that can be beneficial when it occurs normally or harmful when the cell “eats” itself in excessive autophagy. This underscores the connection between autophagy and programmed cell death as well as necrosis. It also relates to the role of autophagy in stroke, where it may exert either a protective or devastating effect depending on the duration of stroke-related stress and whether it occurs in the ischemic or reperfusion phase [119].

Exosomes derived from mesenchymal stem cells (MSCs) have been demonstrated to protect the brain from the effects associated with stroke [120]. It was demonstrated that exosomes derived from bone MSCs obtained from the mouse MCAO model of stroke in vivo and from an in vitro BV-2 cell model of oxygen and glucose deprivation/reoxygenation (OGD/RX) exhibited neuroprotective effects, as indicated by increased cell viability and reduced apoptosis in OGD/RX-affected BV-2 cells [121]. This effect was mediated by the involvement of long noncoding RNA KLF3 antisense RNA 1 (KLF3-AS1), which inhibited apoptosis by increasing autophagy. Furthermore, KLF3-AS1 was found to recruit ETS variant transcription factor 4 (ETV4), which upregulated the expression of SIRT1. Therefore, autophagy may be promoted to alleviate ischemia/reperfusion injury via the ETV4/SIRT1 axis. Using the same in vivo model systems, it was shown that ubiquitin-specific peptidase 18 (USP18) and fat mass and obesity-associated protein (FTO) expression decreased after OGD/R [113]. Dysfunctional mitochondria and apoptosis in OGD/R-stimulated cells were eradicated by USP18/FTO overexpression through the activation of mitophagy. Upregulation of FTO restrained m6A modification of SIRT6, increasing its expression and the subsequent activation of 5′-AMP-activated protein kinase catalytic subunit alpha-1 (AMPK)/PGC-1α/RAC-alpha serine/threonine-protein kinase (AKT) signaling. Therefore, SIRT6 may exert its protective effect through its primary function, promoting mitophagy and stimulating the AMPK/PGC-1α/AKT axis with the involvement of USP18. Another ubiquitin-specific peptidase, USP7, has been shown to promote PINK1/Parkin-dependent mitophagy, thereby alleviating cerebral ischemia-reperfusion injury by deubiquitinating and stabilizing SIRT1 [122].

The role of SIRT1-mediated autophagy in stroke was also confirmed in patients with stroke [123]. An increased expression of SIRT1 and proteins essential for autophagy, including Beclin 1, autophagy-related proteins ATG3, ATG5, ATG7, ATG12-5, and LC3-II/I, was observed postmortem in the brains of stroke patients. Increased immunoreactivity of SIRT1, Beclin 1, ATG7, LC3-I/II, and cleaved caspase-3 was also noted in stroke brains. Therefore, SIRT1 can be upregulated in stroke alongside genes encoding essential autophagy proteins and caspase-3, which is important in apoptosis.

SIRT1 was accumulated through the action of Angong Niuhuang Pill (ANP), a natural substance reported for its neuroprotective effects [124]. Acetylomics and proteomics results suggested that ANP regulated autophagy at the transcriptional level by modulating H4K16ac in the mouse MCAO. Moreover, ANP pretreatment reduced H4K16ac levels, decreased LC3B-II/I ratios, upregulated sequestome 1 (SQSTM1/p62), and suppressed the expression of ATG5 and ATG7. The ability of the SIRT1 inhibitor, EX527, to counteract these effects underscored the importance of the SIRT1–H4K16ac pathway in mediating the protective action of ANP against cerebral ischemia-reperfusion injury. Therefore, extensive autophagy triggered by ischemia-reperfusion injury may be prevented by the SIRT1–H4K16ac pathway.

Several earlier reports have linked sirtuins, specifically SIRT1, to stroke and autophagy, reinforcing the importance of this connection in stroke pathogenesis.

### 5.3. Neuroinflammation

Cerebral ischemic injury leads to increased microglial activation, resulting in the release of pro-inflammatory factors that hinder neurological function in the acute phase of a stroke [125]. On the other hand, sirtuins regulate microglial activation and inflammation after brain injury [126]. Therefore, it is justified to explore the role of sirtuins in stroke-related and microglia-mediated neuroinflammation.

It has been shown that SIRT5 expression in microglia increased in the early phase of stroke in MCAO mice [127]. SIRT5 interacted with and desuccinylated annexin A1 (ANXA1), resulting in the hyperactivation of microglia and increased expression of proinflammatory cytokines and chemokines. This effect led to neuronal cell damage after a stroke. Microglia-specific forced overexpression of SIRT5 worsened ischemic brain injury, while the downregulation of SIRT5 exhibited neuroprotective and cognitive-preserving effects against ischemic brain injury. Therefore, the overexpression of SIRT5 can be considered a therapeutic target in stroke.

An increased expression of SIRT1 was observed in a rat MCAO model of stroke that was intraarterially infused with MSCs [128]. These animals showed improvements in behavioral and motor functions, along with reductions in infarct size, average neuronal length, and density. An increased expression of SIRT1 was linked to greater expression of various inflammatory and apoptotic markers. Thus, SIRT1 may mediate the beneficial effects of MSCs in stroke by regulating inflammasome signaling.

Wogonin, a natural flavonoid, reduced infarct size, decreased brain edema, improved neurological deficits, and mitigated histopathological damage in the mouse MCAO and OGD/R models of stroke [129,130]. Wogonin also decreased the activation of microglial cells and inflammatory factors, including tumor necrosis factor alpha (TNFA), interleukins IL1B, IL6, and IL10, in brain tissue and serum after cerebral ischemic/reperfusion injury. It activated the AMPK/SIRT1 signaling pathway and inhibited the upregulation of NOD-like receptor family pyrin domain containing 3 (NLRP3) inflammasome-related molecules. The SIRT3–NLRP3 interaction was shown to be important for protecting neural stem and progenitor cells, and therefore for neurogenesis in ischemically damaged brain [131]. Moreover, the SIRT1/BRC3/NLRP signaling pathway played a role in reducing brain damage through transcutaneous electrical acupoint stimulation by mitigating neuroinflammation and oxidative stress [132].

Apelin-13, an endogenous peptide, improved neurological function after a stroke, reduced infarct volume, decreased cerebral edema, maintained the integrity of the BBB, inhibited neuronal apoptosis, and diminished neuroinflammation by lowering microglial activation in the MCAO model [133]. In the in vitro BV2/OGD model, apelin-13 inhibited the release of pro-inflammatory cytokines induced by OGD and promoted anti-inflammatory responses. It operates by upregulating SIRT1, which subsequently blocks NF-κB signaling and reduces the expression of inflammatory mediators. Therefore, the apelin-13-SIRT1–NF-κB pathway may represent an effective therapeutic strategy to reduce neuroinflammation and enhance stroke recovery. The SIRT1–NF-κB pathway has been shown to alleviate inflammasome signaling and cellular apoptosis in the intra-arterial mesenchymal stem cell therapy following stroke [134]. The SIRT1–FOXO3A signaling pathway was crucial for the neuroprotective effect of bergenin, a C-glycoside of 4-O-methylgallic acid, in the MCAO model of stroke. Similar results were observed for tanshinone IIA, a quinone derived from a Chinese herb [133]. Another compound derived from a natural substance, wagoniside, extracted from Radix Scutellariae, activated the NRF2–SIRT3 signaling pathway in an in vitro model of stroke and reduced mortality rates, neurological deficits, cerebral infarct size, and brain water content in MCAO.

The involvement of sirtuins in stroke-related neuroinflammation may be regulated epigenetically. miRNA-200b-5p negatively regulates SIRT1 translation and is downregulated by myc proto-oncogene protein (MYC, c-myc) in stroke model systems, showing a negative correlation with SIRT1 expression [134]. MYC enhanced neurological function, reduced inflammation and neuronal apoptosis, and diminished brain pathology while promoting neuronal survival in MCAO mice. Therefore, SIRT1 may be a key protective protein in stroke that can be epigenetically regulated. It is not surprising that SIRT1 is subject to epigenetic regulation, but identifying the elements of its regulation in stroke enables targeting them for therapy and prevention of this disease. Another aspect of the epigenetics concerning the sirtuin/stroke connection is the action of the lncRNA SNHG8, which sponges miR-449c-5p and regulates the SIRT1/FOXO1 pathway to inhibit microglia activation and decrease BBB permeability in ischemic stroke.

Figure 3 does not include CSD, as direct experimental evidence linking specific sirtuins to the modulation of CSD in migraine models is still limited and under investigation.

In summary, most studies demonstrate the beneficial effects of sirtuins in stroke; however, this should not be generalized, as some studies report their detrimental influence. It has been shown that SIRT5 promoted ischemia/reperfusion-induced blood–brain barrier damage after stroke [135].

## 6. Conclusions and Perspectives

Migraine and stroke have numerous common aspects, likely reflecting shared mechanisms in their pathogenesis. The common denominator for most, if not all, of these aspects is oxidative stress. Sirtuins are crucial regulators of cellular and organismal responses to oxidative stress. Therefore, they may play a role in the oxidative stress-related elements of the pathogenesis of both conditions. Furthermore, sirtuins can link the pathogenesis of migraine and stroke, contributing to the migraine–stroke connection. However, it is important to emphasize that oxidative stress is not the only connection between sirtuins and migraine/stroke.

Generally, most studies on the role of sirtuins in migraine and stroke pathogenesis report their beneficial effects. However, sirtuins are a diverse family of proteins that exhibit significant variation in their structure, localization, and functionality. Some studies indicate that certain sirtuins may negatively impact brain vascular homeostasis [136].

Due to their fundamental activity as histone deacetylases, sirtuins may influence the expression of any gene and can therefore be important for phenotypic traits related to that gene. However, sirtuin activity is not always necessary for regulating gene expression, which raises questions about the role of sirtuins in the pathogenesis of migraine and stroke.

Several reports suggest that sirtuins are involved in the pathogenesis of hemorrhagic stroke, which is also linked to migraines [137]. Although hemorrhagic stroke and ischemic stroke are still “strokes,” their mechanisms are significantly different. Moreover, headaches associated with intracranial aneurysms, the rupture of which is the most frequent cause of hemorrhagic stroke, may be misdiagnosed as migraines. Therefore, we cannot extend the involvement of sirtuins in ischemic stroke to hemorrhagic stroke.

Based on the information presented in the previous sections, we conclude that sirtuins may link migraine with stroke through their involvement in antioxidant defense, mitochondrial quality control, neuroinflammation, and autophagy (Table 2). These mechanisms are undoubtedly significant in the pathogenesis of various diseases, and their specificity in the migraine–stroke connections lies in their mediators. However, the role of some of these proteins in the pathogenesis of migraine and/or stroke requires further investigation, as it is not well established at present.

The role of autophagy warrants additional commentary. As noted in the previous section, autophagy is a vital mechanism in stroke, and sirtuins may stimulate it to provide a protective effect against the consequences of stroke. While autophagy is not recognized as a mechanism in migraine pathogenesis, the importance of oxidative stress in migraine and the necessity to eliminate RONS-related cellular debris suggest that autophagy may play a significant role in migraine pathogenesis. Furthermore, we contended that impaired autophagy in glial cells might activate secretory autophagy and lead to the release of specific proteins [138]. These proteins may induce autophagy in neurons to eliminate cellular debris caused by oxidative stress, which occurs in the brain as a response to migraine-related energy deficits. Consequently, migraine-related signaling may hinder degradative autophagy while promoting secretory autophagy in microglia and degradative autophagy in neurons. The significance of autophagy in migraine was also emphasized in a recent review [139].

Most studies on the role of sirtuins in migraine and stroke focus on SIRT1 and SIRT2; however, this does not indicate the particular significance of these two proteins. Instead, it reflects the history of discoveries among the sirtuin family, and these findings may not represent other family members. We cited consistent results demonstrating that sirtuins may be involved in the pathogenesis of migraine and stroke by inhibiting oxidative stress/RONS overproduction, enhancing DDR, and improving autophagy and cellular metabolism. Nevertheless, many other proteins and non-protein compounds may also exert such beneficial effects [140].

The connection between migraines and strokes should also be analyzed from an aging perspective. Migraines primarily affect young and middle-aged individuals, while the highest prevalence of strokes occurs in the elderly. Therefore, the role of sirtuins in the correlation between migraines and strokes should be examined from this viewpoint, especially since oxidative stress is a crucial factor in the age-related prevalence of these diseases [10].

One of the fundamental questions regarding the migraine–stroke connection, specifically concerning the role of sirtuins, is whether these interrelationships have any therapeutic implications. Prophylactic treatment aimed at the ischemic origin of secondary aura may prevent both migraine and stroke. Therapeutic strategies that target the shared elements of migraine and stroke pathophysiology may have the potential to alleviate migraine symptoms and aura effects while preventing stroke. As we have argued in this work, sirtuins may represent one such common element. Further studies are required to explore the significant therapeutic potential of sirtuins, as they are relatively novel entities—the human SIRT1 gene was cloned at the turn of the century and characterized in 2006 [141].

## Figures and Tables

**Figure 1 ijms-26-06634-f001:**
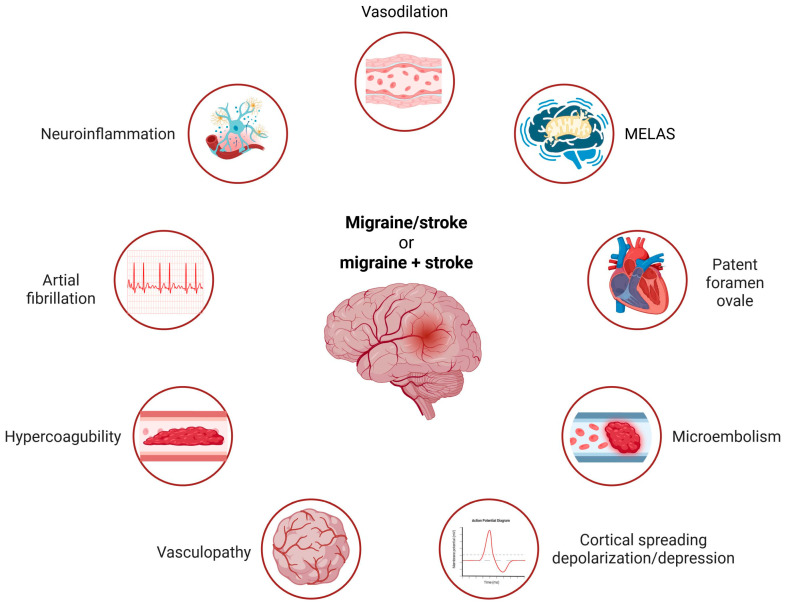
The main common risk factors for migraine and stroke include several recognized contributors. Smoking, obesity, and high blood pressure are noted, but these factors are not very specific to either disease. Conversely, atrial fibrillation, hypercoagulability, and microembolism are weak risk factors for migraine alone but are strongly associated with the migraine–stroke connection. MELAS—mitochondrial encephalomyopathy, lactic acidosis, and stroke-like episodes—is representative of other mitochondrial diseases associated with migraine and stroke. The common denominator for these risk factors is oxidative stress, with some factors being relevant only to migraines with aura. Created in https://BioRender.com.

**Figure 2 ijms-26-06634-f002:**
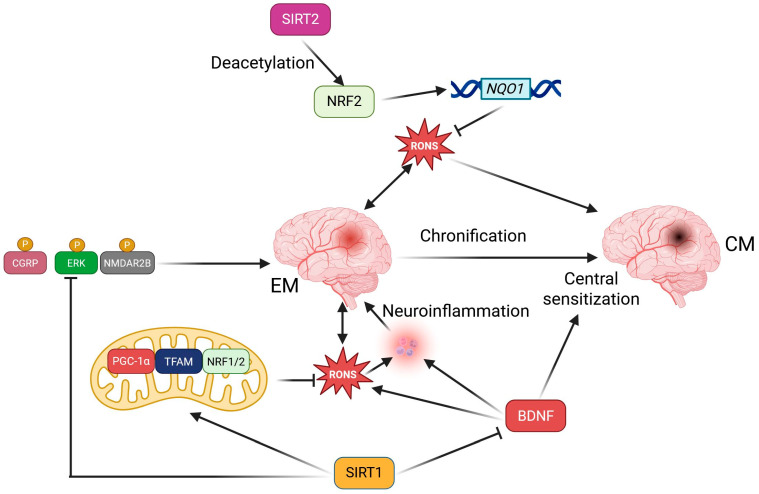
Sirtuins 1 and 2, SIRT1 and SIRT2, may help prevent the induction, progression, and chronification of migraines in several ways, primarily by indirectly modulating oxidative stress that may arise from excessive energy production in brain mitochondria and various migraine triggers. Detailed descriptions and definitions of abbreviations are provided in the main text.

**Figure 3 ijms-26-06634-f003:**
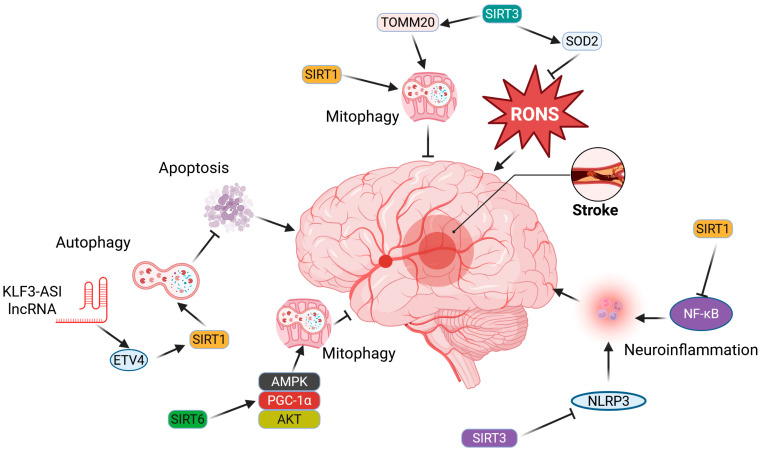
Sirtuins (SIRTs) can play significant roles in both stroke promotion and inhibition, with only a few representative mechanisms highlighted here. SIRT1 may inhibit stroke-promoting processes, including oxidative stress, apoptosis, and neuroinflammation, by reducing the levels of reactive oxygen and nitrogen species (RONS), activating autophagy/mitophagy, and blocking the signaling of proteins essential for inflammation, such as nuclear factor NF-kappa-B (NF-κB) and NOD-like receptor family pyrin domain-containing 3 (NLRP3). SIRT6 may enhance signaling pathways associated with antioxidant defense and mitochondrial quality control, particularly mitophagy. Detailed descriptions and definitions of abbreviations can be found in the main text.

**Table 1 ijms-26-06634-t001:** Main subcellular localization, types of biochemical reactions, and primary targets for sirtuins.

Sirtuin	Main Localization	Key Activities	Main Targets	References
SIRT1	Nucleus	Deacetylation	H3 *, H4, FOXO, NF-κB, HIF-1α, XRCC6, WRN	[50,51]
SIRT2	Cytoplasm	Deacetylation, deacylation	p53, FOXO	[52,53]
SIRT3	Mitochondria	Deacetylation	AceCS2	[54]
SIRT4	Mitochondria	ADP-ribosylation	GDH	[55]
SIRT5	Mitochondria	Desuccinylation, demanlonylation	CPS1	[56]
SIRT6	Nucleus	Deacetylation, deacylation	H3, H4, FOXO, HIF-1α, PARP1	[57,58]
SIRT7	Nucleous	Deacetylation	H3, H4	[59,60]

* Abbreviations: AceCS2, acetyl-CoA synthetase 2; CPS1, carbamoyl phosphate synthetase; FOXO, forkhead box O; GDH, glutamate dehydrogenase; H3/4, histones H3 and H4; HIF-1α, hypoxia-inducible factor 1-alpha; NF-κB, nuclear factor kappa-light-chain-enhancer of activated B cells; p53, cellular tumor antigen p53; PARP1, poly (ADP-ribose) polymerase 1; WRN, Werner syndrome ATP-dependent helicase; XRCC6, X-ray repair cross-complementing protein 6.

**Table 2 ijms-26-06634-t002:** Main mechanisms that underlie the involvement of sirtuins in migraine–stroke connections.

Sirtuin	Mechanism	Mediators
SIRT1	Antioxidative effects	PGC-1α *, ERK, NMDAR2B
	Mitochondrial quality control	PGC-1α, NRF1/2, TFAM, DRP1
	Neuroinflammation	miR-155-5p, NF-κB, TNFA, IL1B, IL6, IL10, AMPK, NLRP3, apelin-13, miRNA-200b-5p, MYC, lncRNA SNHG8, miR-449c-5p
	Autophagy	lncRNA KLF3 antisense RNA 1, ETV4, ATG5, ATG7, Beclin 1, LC3-II/I, AQSTM1/p62
	Mitophagy	PINK1/Parkin
SIRT2	Antioxidant effects	NRF2, NQO1
SIRT3	Antioxidant effects	SOD2, FOXO3A, NRF2
	Mitochondrial quality control	TOMM20
	Neuroinflammation	NLRP3, BRC3
SIRT4	Antioxidant effects	SOD2, FOXO3A
SIRT5	Neuroinflammation	ANXA1
SIRT6	Mitophagy	FTO, USP18, AMPK, PGC-1α, AKT
SIRT7	Mitochondrial quality control	PGC-1α, NRF1/2, TFAM

* Abbreviations are defined in the main text.

## Data Availability

Not applicable.

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
