# Peer review of "Sirtuins Contribute to the Migraine–Stroke Connection"

_ijms, 2025, doi:10.3390/ijms26146634_

Round 1

Reviewer 1 Report

Comments and Suggestions for Authors

Overall, in this review the authors provided description on migraine pathogenesis, rather brief description of sirtuins (SIRTs) biology, rather brier description of stroke pathogenesis, and tried to make a case of involvement of SIRTs in migraine and stroke.

Please find below my comments and suggestions.

In the abstract the authors state “Sirtuins (SIRTs) is a six member family <>”. This is not correct. It is a seven member family.

What about prevalence of stroke in patients without any type of migraine (MO, MA)? Could you please provide such data?

Gene names as well as nucleotide variants should be italicised (for example MTL1 gene, m.3243A>G variant). All abbreviations should be introduced upon first use and only once, which is not always the case in the current version of the manuscript.

I also suggest replacing word “mutation/mutations” with “pathogenic nucleotide variant/causal nucleotide variant”.

As for SIRT7, I would say that it has predominantly nucleolar localisation, which in a way makes it a unique member of SIRT family.

Regarding the Table 1. I recommend either removing info about role of the given SIRTs in several diseases (as it’s definitely not a complete list of diseases/pathological conditions in which SIRTs play role) , or significantly expanding it, to provide as much info as possible. Some statements are too broad/general, for example “brain functions” or “metabolism” for SIRT2. Indeed, any protein which is present in brain cells is related to “brain function”. Please by more specific. Further, I find info in this Table a bit misleading, for example, time and time again authors mention cancer in this Table, but not diabetes or several neurological conditions (Alzheimer’s disease, Parkinson’s disease, Huntington’s disease), as if SIRTs were not involved in their pathogenesis (but they are involved, and authors discuss it further in the text!). How is “Gene expression regulation via deacetylation of chromatin proteins and transcription factors” (listed as a function of SIRT1) different to “Histone deacetylation, regulation of transcriptional co-activators” (SIRT2) or “Gene expression” (for SIRT6, SIRT7)? Do the authors want to say that only SIRT1, SIRT2, SIRT6, SIRT7 have histone-deacetylase activity?

Instead, the authors could have summarised main types of biochemical reactions in which SIRTs are involved and main targets of SIRTs. (Information about sub-cellular localisation of SIRTs also seems to be useful).

The authors state “Emerging evidence demonstrates an important role in the pathogenesis of various diseases, including cardiovascular disease, cancer, neurodegenerative diseases, epilepsy, liver diseases, premature aging, and other serious syndromes [49].” An important role of what? The sentence is not complete.

The authors state “SIRT4 has been shown to generate RONS”. It’s not correct way to say that increased levels of SIRT4 are associated with elevated levels of RONS.

The authors state “numerous migraine triggers are directly or indirectly associated with oxidative stress”. Please provide more info regarding these triggers.

What about interplay of SIRT7 and mitochondria? Indeed, to have significant impact on mitochondria there is no need to be localised in mitochondria, the impact might be crucial but indirect.

If, as authors and others suggest, migraine might by linked to defects of antioxidant system, such as catalase dysfunction, perhaps prevalence of migraine is higher in other conditions linked to catalase disfunction? Are there such studies to add to discussion?

The sub-chapter about BDNS seems to be irrelevant to the subject of study – I do not see a strong link abetween BDNF and SIRTs in a given context.

The authors state that “SIRT1’s inhibitor SRT1720” alleviates microglia activation caused by miR-155. But SRT1720 is not an inhibitor, its an activator of SIRT. Also, does it activate other SIRTs?

Is superoxide dismutase 2 (SOD2) a substrate of SIRT3 (in other words, does SIRT3 deacetylates SOD2) as the authors claim? (I was under impression that SOD2 expression is regulated by FOXO3A, which is target of SIRT3. Please clarify.

What do you think about translational value (chances to be used in clinic) of SIRT activators?

Regarding Table 2, entitled “Mechanisms that underlie the involvement of sirtuins in migraine-stroke connections”. In the first line the authors mention BDNF as one of such mediators, although nothing is said about its involvement in stroke. I recommend removing both Tables from the text as in my opinion they raise questions rather than provide clarification.

Reviewer 2 Report

Comments and Suggestions for Authors

This narrative review effectively explores the shared pathophysiological mechanisms between migraine and stroke, emphasizing the role of sirtuins (SIRTs) as potential molecular links. By highlighting their antioxidant properties, mitochondrial localization, and involvement in neuroinflammation, autophagy, and mitochondrial quality control, the authors present a compelling case for sirtuins as therapeutic targets. The review also underscores the diagnostic challenges posed by overlapping symptoms, such as migraine with aura mimicking stroke, and calls for further research into mediators that could improve clinical outcomes through targeted interventions. This review paper presents a highly compelling topic and is well-supported by diverse literature, offering valuable insights into the molecular links between migraine and stroke. My suggestion is as follow.

  1. Although the prevalence of stroke associated with migraine is not significant in the general population or among men, it is notably higher in women. This suggests that sex may act as a mediator in the migraine-stroke relationship or that sex-specific pathophysiological mechanisms are involved. Therefore, it is recommended to include a separate section and a well-organized figure illustrating the sex-based differences in the migraine-stroke connection.
  2. In Figure 1, while common risk factors are presented, literature indicates that certain factors—such as smoking and obesity—have a stronger impact. Adding visual cues to differentiate the strength of these risk factors would enhance the figure’s clarity and help readers intuitively grasp the content. Additionally, the inclusion of atrial fibrillation, hypercoagulability, and microemboli as migraine risk factors is confusing, as their mechanisms in triggering migraine are unclear. These aspects should be further elaborated in the main text.
  3. It appears that shared risk factors for migraine and stroke may influence both conditions through common pathways such as cortical spreading depression (CSD) and oxidative stress, possibly mediated by brain structural changes or genetic factors. Moreover, migraine itself may elevate stroke risk. If this interpretation is correct, a comprehensive figure is needed to illustrate where and how sirtuins act within this axis.
  4. Are there any studies comparing patients with migraine but no stroke versus those with both migraine and stroke? Identifying clinical differences between these groups could help pinpoint stronger risk factors or mediators that predispose migraine patients to stroke, potentially aiding in prevention strategies.
  5. Have any studies investigated whether the expression of sirtuin genes is associated with prognosis in migrainous stroke? Since managing stroke risk factors often leads to better outcomes post-stroke, understanding the prognostic role of sirtuins could be clinically valuable.
  6. Does the abbreviation "CM" refer to chronic migraine? Also, the abbreviation "IS" seems to represent both "inflammatory soup" and "ischemic stroke," which may cause confusion. It is advisable to clarify and separate these terms to avoid ambiguity.
  7. There are a few minor typographical errors in the text that could be corrected to improve clarity and professionalism.
Comments on the Quality of English Language

There are a few minor typographical errors in the text that could be corrected to improve clarity and professionalism.

Reviewer 3 Report

Comments and Suggestions for Authors

This review article delves into the connection between migraine and stroke, highlighting the role of sirtuins (SIRTs) as potential linking factors. It discusses how sirtuins, a family of NAD+-dependent histone deacetylases, may influence oxidative stress, mitochondrial quality control, neuroinflammation, and autophagy, all of which are implicated in the pathogenesis of both migraine and stroke. The article also emphasizes the therapeutic potential of targeting sirtuins to alleviate migraine and prevent stroke.

  1. Provide a more detailed discussion of the specific functions of each sirtuin family member in the context of migraine and stroke. This would help clarify their individual roles and potential interactions.
  2. While the article mentions oxidative stress as a common mechanism, further exploration of other shared mechanisms such as endothelial dysfunction and vascular remodeling could strengthen the connection between migraine and stroke. For instance, the author can refer to the following paper to elaborate on how the multi-mechanism combined strategy can more rapidly facilitate the treatment of stroke (https://doi.org/10.1016/j.apsb.2025.01.005).
  3. Incorporate the latest research findings on sirtuins and their role in neurological diseases to ensure the review is up-to-date and comprehensive.
  4. Since the article briefly mentions hemorrhagic stroke, a more in-depth analysis of sirtuins' role in this type of stroke would provide a more complete picture of their involvement in cerebrovascular diseases.
  5. Discuss potential therapeutic interventions targeting sirtuins in greater detail, including preclinical and clinical trials, to highlight their translational potential.
  6. Enhance the discussion on how oxidative stress modulates sirtuin activity and vice versa, providing a deeper understanding of their interplay in disease pathogenesis.

Round 2

Reviewer 1 Report

Comments and Suggestions for Authors

The authors have addressed all my comments, and I am glad that my comments and suggestions contributed to the further improvement of the text. 

Reviewer 2 Report

Comments and Suggestions for Authors

The issues of concern have been satisfactorily resolved through the author's response.